# A FOUNDATION MODEL FOR ERROR CORRECTION CODES

**Yoni Choukroun**
The Blavatnik School of Computer Science
Tel Aviv University
choukroun.yoni@gmail.com

**Lior Wolf**
The Blavatnik School of Computer Science
Tel Aviv University
wolf@cs.tau.ac.il

## ABSTRACT

In recent years, Artificial Intelligence has undergone a paradigm shift with the rise of foundation models, which are trained on large amounts of data, typically in a self-supervised way, and can then be adapted to a wide range of downstream tasks. In this work, we propose the first foundation model for Error Correction Codes. This model is trained on multiple codes and can then be applied to an unseen code. To enable this, we extend the Transformer architecture in multiple ways: (1) a code-invariant initial embedding, which is also position- and length-invariant, (2) a learned modulation of the attention maps that is conditioned on the Tanner graph, and (3) a length-invariant code-aware noise prediction module that is based on the parity-check matrix. The proposed architecture is trained on multiple short- and medium-length codes and is able to generalize to unseen codes. Its performance on these codes matches and even outperforms the state of the art, despite having a smaller capacity than the leading code-specific transformers. The suggested framework therefore demonstrates, for the first time, the benefits of learning a universal decoder rather than a decoder optimized for a given code.

## 1 INTRODUCTION

Reliable digital communication relies on the design of codes that can be accurately decoded when transmitted over noisy channels. The optimal decoding is defined by the NP-hard maximum likelihood rule, and the efficient decoding of codes remains an open problem.

Recently, powerful learning-based decoders have been introduced, borrowing from the architecture of well-proven deep models. For example, a Transformer-based decoder that incorporates the error correction code (ECC) into the architecture has been recently proposed by Choukroun & Wolf (2022a), outperforming existing methods by sizable margins and at a fraction of their time complexity. This architecture has been subsequently integrated into a denoising diffusion models paradigm, further improving results (Choukroun & Wolf, 2022b).

A major drawback of current ECC neural decoders is the dedicated adaptation of the architecture and optimization of the model with respect to each single code of interest. This major challenge prevents a trained neural decoder from being employed for the decoding or fine-tuning of other codes or settings, even for the same family of codes and settings but with different lengths.

Large models known as foundation models (Bommasani et al., 2021) have revolutionized deep learning by addressing multiple complex downstream tasks that pose a challenge to alternative methods. These models are typically initiated through training on extensive unlabeled datasets, primarily by employing self-supervised pretext tasks. Subsequently, transfer learning to new tasks utilizes strategies such as zero-shot learning, prompt-engineering, or fine-tuning (Brown et al., 2020).

In this work, we consider the design and training of a foundation neural decoder for error correction codes based on the Transformer architecture of Vaswani et al. (2017), which is capable of adapting and generalizing to any code. As far as we can ascertain, this is the first time a universal neural decoder has been presented. Beyond the conceptual novelty, we make three technical contributions: (i) we adapt the Transformer input embedding in order to remain *invariant* to code length (ii) the positional embedding, as well as the code, are added as *relative* positional encoding that is integrated

into the self-attention via a *learned* mapping of the node distances in the Tanner graph, and (iii) a size-invariant prediction module that is conditioned on the parity-check matrix and is based on a learned aggregation function. Applied to a wide variety of codes, our method is able to reach and even surpass the performance of state-of-the-art per-code learning-based solutions on the zero-shot, pretrained, and fine-tuned settings. This is obtained despite remaining entirely code-invariant and employing an extremely shallow architecture.

## 2 RELATED WORKS

Previous work on neural decoders was divided by Raviv et al. (2020) into two main classes: model-based or model-free. Model-based decoders implement parameterized versions of classical Belief Propagation (BP) decoders, where the Tanner graph is unfolded into a neural network in which weights are assigned to each variable edge. This results in an improvement in comparison to the baseline BP method for short codes (Nachmani et al., 2016; Nachmani & Wolf, 2019; 2021). Model-based decoders benefit from a strong theoretical background, but the architecture is overly restrictive.

Model-free decoders employ general types of neural network architectures. Earlier approaches (Cammerer et al., 2017; Gruber et al., 2017; Kim et al., 2018) employed stacked fully connected networks or recurrent neural networks that have difficulties in learning the code. However, some of the training approaches are still in use. For example, our work employs the preprocessing of Bennatan et al. (2018), which transforms the channel output such that the decoder remains provably invariant to the codeword. This allows the training of generic neural decoders without the risk of codeword overfitting. Choukroun & Wolf (2022a) introduced the Error Correction Code Transformer (ECCT), obtaining SOTA performance. The model embeds the signal elements into a high-dimensional space where the analysis is more efficient, while the information about the code is integrated via a masked self-attention mechanism. Subsequently, Choukroun & Wolf (2022b) extended the denoising diffusion paradigm to ECC, iteratively applying an ECCT, further improving the SOTA by a large margin.

We note that neural decoder contributions generally focus on short and moderate-length codes for two main reasons: (i) classical decoders are proven to reach the capacity of the channel for large codes, preventing any potential enhancement, and (ii) the emergence of applications driven by the Internet of Things created the requirement for optimal decoders of short to moderate codes. For example, 5G Polar codes have code lengths of 32 to 1024 (ESTI, 2021).

Recently, the ML community has been focusing intensively on large foundation models trained on internet-scale datasets, which achieve state-of-the-art performance on a diverse range of learning tasks. Rather than learning task-specific models from scratch, the foundation models are adapted via fine-tuning or few-shot/zero-shot learning strategies and subsequently deployed on a wide range of domains (Brown et al., 2020; Radford et al., 2021). Such foundation models enable the transfer and sharing of knowledge across domains, mitigate the need for task-specific training data, and have been applied to several fields, such as Natural Language Processing (LLM)(Wei et al., 2022; Touvron et al., 2023; Brown et al., 2020), computer vision (Saharia et al., 2022; Rombach et al., 2022; Ramesh et al., 2022), multimodal analysis (Radford et al., 2021; Li et al., 2022; Awadalla et al., 2023; OpenAI, 2023), and even reinforcement learning (Reed et al., 2022).

## 3 PROBLEM SETTING AND BACKGROUND

In this work, we assume a standard transmission protocol using a linear code $C$. The code is defined by a generator matrix $G \in \{0,1\}^{k \times n}$ and the parity check matrix $H \in \{0,1\}^{(n-k) \times n}$ defined such that $GH^T = 0$ over the order 2 Galois field $GF(2)$. The parity check matrix $H$ entails what is known as a Tanner graph, which consists of $n$ variable nodes and $(n-k)$ check nodes. The edges of this graph correspond to the on-bits in each column of the matrix $H$.

The input message $m \in \{0,1\}^k$ is encoded by $G$ to a codeword $x \in C \subset \{0,1\}^n$ satisfying $Hx = 0$ and transmitted via a Binary-Input Symmetric-Output channel, e.g., an AWGN channel. Let $y$ denote the channel output represented as $y = x_s + \varepsilon$, where $x_s$ denotes the Binary Phase Shift Keying (BPSK) modulation of $x$ (i.e., over $\{\pm 1\}$), and $\varepsilon$ is a random noise independent of

the transmitted $x$. The main goal of the decoder $f : \mathbb{R}^n \to \mathbb{R}^n$ is to provide a soft approximation $\hat{x} = f(y)$ of the codeword.

We follow the preprocessing of Bennatan et al. (2018); Choukroun & Wolf (2022a), in order to remain provably invariant to the transmitted codeword and to avoid overfitting. The preprocessing transforms $y$ to a $2n - k$ dimensional codeword invariant vector defined as $\tilde{y} = h(y) = [|y|, s(y)]$, where, $[\cdot, \cdot]$ denotes vector concatenation, $|y|$ denotes the absolute value (magnitude) of $y$ and $s(y) \in \{0, 1\}^{n-k}$ denotes the binary code *syndrome*. The syndrome is obtained via the $GF(2)$ multiplication of the binary mapping of $y$ with the parity check matrix such that

$$s(y) = Hy_b := H\text{bin}(y) := H\big(0.5(1 - \text{sign}(y))\big). \tag{1}$$

The induced parameterized decoder $f_\theta : \mathbb{R}^{2n-k} \to \mathbb{R}^n$ with parameters $\theta$ aims to predict the multiplicative noise denoted as $\tilde{\varepsilon}$ and defined such that $y = x_s \odot \tilde{\varepsilon}$. The prediction of the multiplicative noise instead of the additive physical one is used in order to remain invariant to the transmitted codeword, since the syndrome is codeword-invariant and $|y| = |x_s \tilde{\varepsilon}| = |\tilde{\varepsilon}|$, thereby avoiding the risk of code overfitting, as described by Bennatan et al. (2018) and the proof of lemma 1 of (Richardson & Urbanke, 2001). The final prediction takes the form $\hat{x}_s = \text{sign}(y \odot f_\theta(|y|, Hy_b))$. An illustration of the coding procedure under the foundation model setting is given in Appendix A.

## 3.1 Error Correction Code Transformer

The state-of-the-art ECCT (Choukroun & Wolf, 2022a) has been recently proposed for neural error decoding. It consists of a transformer architecture (Vaswani et al., 2017) with several modifications. Following Bennatan et al. (2018), the model's input $h(y)$ is defined by the concatenation of the codeword-independent magnitude and syndrome, such that $h(y) := [|y|, 1 - 2s(y)] \in \mathbb{R}^{2n-k}$. Each element is then embedded into a high-dimensional space for more expressivity, such that the initial positional embedding $\Phi \in \mathbb{R}^{(2n-k) \times d}$ is given by $\Phi = \big(h(y) \cdot 1_d^T\big) \odot W$ where $W \in \mathbb{R}^{(2n-k) \times d}$ is the learnable embedding matrix and $\odot$ is the Hadamard product.

The interaction between the bits is performed naturally via the self-attention modules coupled with a binary mask derived from the parity-check matrix in order to integrate information about the code

$$A_H(Q, K, V) = \text{Softmax}(d^{-1/2}(QK^T + g(H)))V, \tag{2}$$

where $g(H)$ is a fixed binary masking function designed according to the parity-check matrix $H$, and $Q, K, V$ are the classical self-attention projection matrices.

The masking $g(H) : \{0, 1\}^{(n-k) \times n} \to \{-\infty, 0\}^{(2n-k) \times (2n-k)}$ is defined by the adjacency matrix of the Tanner graph extended to two-ring connectivity: the mask's on-bits exist between adjacent nodes (distance one) and secondary neighbors (distance two).

Finally, the transformed embedding is projected onto a one-dimensional vector for the noise prediction with a linear layer, such that with $W_o \in \mathbb{R}^{(2n-k) \times n}$ and $W_{d \to 1} \in \mathbb{R}^{d \times 1}$, we have

$$\hat{\tilde{\varepsilon}} = W_o^T(W_{d \to 1}\Phi) \tag{3}$$

# 4 The ECC Foundation Model

We present the elements of the proposed Foundation Error Correction Code Transformer (FECCT) decoder, the complete architecture, and the training procedure

## 4.1 Code-Invariant Initial Embedding

Traditional Transformers encode each token according to the model's vocabulary embedding; the positional embedding is added at a later stage. In ECCT, a unique model is crafted for every code and length. The initial embedding is designed such that each input bit possesses its distinct embedding vector, providing, as a byproduct, a learned positional encoding.

In our length-invariant context, we propose a new code-invariant embedding, where a single embedding is given for all magnitude elements, and two embeddings are given for every element of the binary syndrome.

Figure 1: For the Hamming(7,4) code: (a) the parity-check matrix, (b) the induced Tanner graph, (c) the ECCT binary masking, (d) the distance matrix $\mathcal{G}(H)$.

Formally, by considering each dimension of $\{\tilde{y}_i\}_{i=1}^{2n-k}$ separately, we define the projection of each element to a high, $d$ dimensional embedding $\{\phi_i\}_{i=1}^{2n-k} \in \mathbb{R}^d$ as:

$$\phi_i = \begin{cases} |y_i| W_M, & \text{if } i \le n \\ W_{(s(y))_{i-n+1}}^S, & \text{otherwise} \end{cases} \tag{4}$$

where $\{W_M, W_0^S, W_1^S\} \in \mathbb{R}^d$ denote the magnitude encoding and one-hot encoding of the binary syndrome elements, respectively, and $(s(y))_j$ denotes the $j$ element of the syndrome vector. Note that unlike ECCT, in which $2n - k$ different embeddings are learned (for a specific code), FECCT learns only three embedding vectors (which are used for all codes). This substantially reduces the number of learned parameters.

Since ECCT has a different embedding for each bit, this fixed embedding is capable of capturing the bit position. In contrast, our method enables the embedding of arbitrary codes of any length, at the price of losing the positional encoding. This is addressed next, using a code-conditioned modulation of the self-attention maps.

## 4.2 TANNER GRAPH DISTANCE MASKING AS CODE AND POSITIONAL ENCODING

For a given algebraic code defined by the matrix $H$, ECCT integrates its information by designing a binary masking function, thresholding the attention between nodes with distances greater than two on the Tanner graph.

In FECCT, the masking serves two purposes. First, similar to ECCT, it integrates the code structure into the transformer. Second, since, unlike ECCT, the initial embedding is position-invariant, the learned masking adds the *relative* position information to the processed elements.

Based on the notion that the Tanner graph captures the relations between every two bits in the code, we consider the distance matrix $\mathcal{G}(H) \in \mathbb{N}^{(2n-k)\times(2n-k)}$, which is constructed from the Tanner graph of the parity-check matrix $H$. Specifically, each element $(i, j)$ in this matrix is defined as the length of the shortest path in the Tanner graph between node $i$ and node $j$. An illustration of $\mathcal{G}(H)$ on the Hamming(7,4) code is presented in Figure 1(d).

In order to incorporate the positional information into the self-attention mechanism, we learn a parameterized mapping $\psi : \mathbb{N} \to \mathbb{R}$ from the elements of the distance matrix $\mathcal{G}(H)$ to a value that is used to modulate the self-attention maps:

$$A_H(Q, K, V) = \left(\text{Softmax}\left(\frac{QK^T}{\sqrt{d}}\right) \odot \psi\big(\mathcal{G}(H)\big)\right)V. \tag{5}$$

This $H$-dependent attention mechanism generalizes that of Eq. 2, where the latter may be approximated as a special case in which $\psi((\mathcal{G}(H)_i)) = \mathbb{1}_{(\mathcal{G}(H))_i \le 2}$ is applied in a non-differentiable manner. The general form of modulation in Eq. 5 captures more information from the Tanner graph, and, therefore, about the code. This allows us, for example, to reconstruct the entire graph from the values used to modulate the self-attention maps, whereas ECCT's binary thresholding function is often not informative enough for reconstruction, as shown in Appendix B.

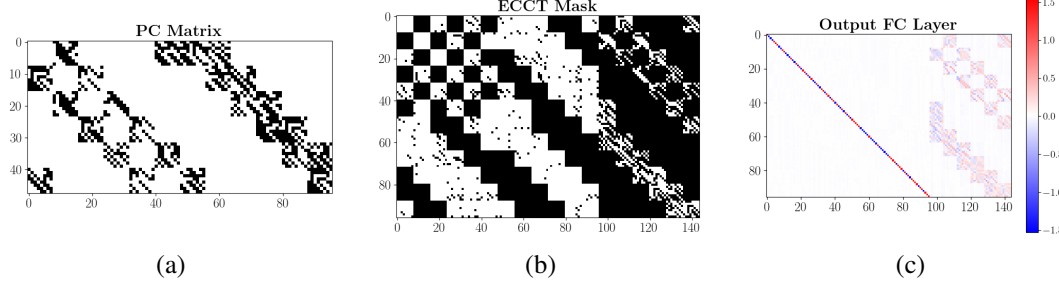

Figure 2: For the LDPC(96,48) code: (a) the parity-check matrix, (b) the cropped ECCT binary masking, (c) the learned fully connected output layer. One can observe that the small dynamic range reduces the fully connected dot-product to a sum of few signed elements

## 4.3 Parity-Check Aware prediction

After the initial encoding, the $(2n-k)$ embeddings are propagated through multiple normalized self-attention and feed-forward blocks towards the output module to provide the final noise prediction.

To obtain the prediction, ECCT makes use of two fully-connected layers, as described in Eq. 3. The first layer $W_{d\to1}$ shrinks each of the $(2n-k)$ embedding vectors from $d$ to 1 and the second $W_o$ reduces the output from $\tilde{y}$'s dimensions $(2n-k)$ to $y$'s dimensions $(n)$.

As can be seen in Fig. 2(c), the learned $W_o$ is sparse and each bit of the prediction is a linear combination of the corresponding bits in $\tilde{y}$ (the diagonal line in the figure) and of its related elements according to the parity-check matrix or induced mask (see the $(n-k)$ rightmost columns). This behavior is not enforced; it emerges organically during training.

Motivated by this phenomenon, in FECCT, we explicitly enforce a similar dependency structure. Since the FECCT is required to be code- and length-invariant, it cannot use a fully connected layer, and this structure is obtained by directly considering the parity check matrix as follows.

We first transform the magnitude and the syndrome parts of the learned embedding separately. The syndrome part is resized from $(n-k)$ embeddings to $n$ embeddings according to an aggregation induced by the parity-check matrix. Then, the $n$ magnitude embeddings are added to the transformed syndrome elements. Finally, the $d$-dimensional embedding is shrunk to one dimension for the prediction. This way, we are able to incorporate the residual information of the syndrome embeddings into the final prediction bits.

Specifically, let us denote the final Transformer's embedding $\phi_o \in \mathbb{R}^{(2n-k)\times d}$, and $W_S, W_M \in \mathbb{R}^{d\times d}$ two learnable affine transforms matrices. We define by $\phi_{o[1:n]} = \phi_{o,M} \in \mathbb{R}^{n\times d}$ and $\phi_{o[n+1:2n-k]} = \phi_{o,S} \in \mathbb{R}^{(n-k)\times d}$ the magnitude and syndrome parts of the embedding, respectively. The output module performs the following projections

$$\hat{\hat{\varepsilon}} = \left(\phi_{o,M}W_M + H^T(\phi_{o,S}W_S)\right)W_{d\to1} \tag{6}$$

where $\hat{\hat{\varepsilon}}$ denotes the predicted noise, and $W_{d\to1}$ denotes the final embedding shrinkage. The proposed aggregation provides both (i) code-awareness in the sense that the aggregation is induced by the parity-check matrix, and (ii) code-invariance in the sense that the aggregation is invariant to code size and can be performed with any code.

## 4.4 Architecture and Training

An illustration of the entire model is given in Figure 3. The initial encoding is performed with a $d$ dimensional one-hot encoding for the syndrome part and a single $d$-dimensional vector for the magnitude part, for a total of three $d$-dimensional parameters. The decoder is defined as a concatenation of $N = 6$ decoding layers composed of self-attention and feed-forward layers interleaved with normalization layers with $d = 128$. The distance embedding is a learned integer-to-scalar mapping, for mapping values $1, \ldots, 10$, where 10 is the maximal distance encountered in a Tanner graph. See

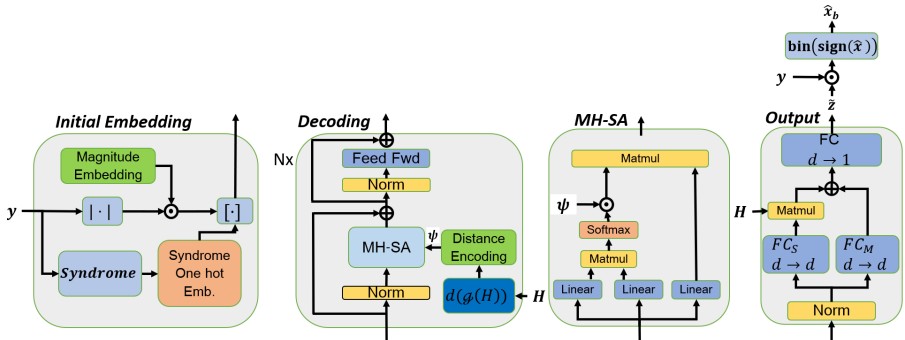

Figure 3: Illustration of the proposed Foundation Transformer architecture.

Appendix C for the histogram of distances in the training dataset. Note that self-attention is turned off by mapping the diagonal (distance of zero) to infinity.

The dimension of the feed-forward network of the transformer is four times that of the embedding $d$, following Vaswani et al. (2017), and is composed of GEGLU layers (Shazeer, 2020), with layer normalization set to the pre-layer setting, as in Klein et al. (2017); Xiong et al. (2020). An eight-head self-attention module is used in all experiments. We note that while larger architectures and longer training times would enable better performance, deepening the accuracy gap from other methods (e.g., GPT-3 (Brown et al., 2020) operates successfully on $2K$ inputs with a similar Transformer model, but with $N = 96, d = 12K$), error correction requires rather light and shallow models that can be deployed on edge devices. The output module is described fully in the previous section.

The training objective is the cross-entropy function, with the goal of learning to predict the *multiplicative* noise $\tilde{\varepsilon}$ (Bennatan et al., 2018). Denoting the *soft* multiplicative noise as $\tilde{\varepsilon}_s$, such that $y = x_s \odot \tilde{\varepsilon}_s$, we obtain $\tilde{\varepsilon}_s = \tilde{\varepsilon}_s \odot x_s^2 = y \odot x_s$. Thus, the binary multiplicative noise to be predicted is defined by $\tilde{\varepsilon} = \mathrm{bin}(y \odot x_s)$, such that the loss computed for a single received word $y$ is

$$\mathcal{L} = -\sum_{i=1}^{n} \tilde{\varepsilon}_i \log(f_\theta(y)) + (1 - \tilde{\varepsilon}_i) \log(1 - f_\theta(y)). \tag{7}$$

The estimated hard-decoded codeword is straightforwardly obtained as $\hat{x}_b = \mathrm{bin}(\mathrm{sign}(f_\theta(y) \odot y))$.

The Adam optimizer (Kingma & Ba, 2014) is used with 512 samples per minibatch, for 3000 epochs, with 1000 minibatches per epoch. We note that while using more epochs can improve performance as demonstrated with large foundation models, the current shallow setting already reaches SOTA performance. We initialized the learning rate to $10^{-4}$ coupled with a cosine decay scheduler down to $10^{-6}$ at the end of the training. No warmup was employed (Xiong et al., 2020).

The code database from Helmling et al. (2019) was web-scraped in order to extract all possible binary codes. Due to our modest computational resources, we limited our training to codes with lengths no longer than 150. A full description of the codes used for training and testing as well as general statistics is given in Appendix C. It is important to note that the training set is highly unbalanced between code families.

Due to the construction of the model and its input preprocessing, the zero codeword is sufficient for training for every code. The additive Gaussian noise is sampled randomly per batch in the $\{2, \ldots, 8\}$ normalized SNR (i.e. $E_b/N_0$) range. Other types of noise can also be used to increase the generalization to noise (e.g., Rayleigh channel), but this is beyond the scope of this paper. Each training batch is constructed as the concatenation of randomly sampled code and noise, with adapted masking of the different elements (e.g., $y$, self-attention mask, distances, etc.).

Training time is around 12 days, without any coding optimization of the self-attention mechanism. The acceleration of the proposed method (e.g. pruning, quantization, distillation, low-rank approximation) (Wang et al., 2020; Lin et al., 2021) is beyond the scope of this paper and is left for future work. Training and experiments are performed on four 12GB GeForce RTX 2080 Ti GPUs. The training time is approximately 320 seconds per epoch. Testing time depends on the code, ranging from 10ms to 30ms per sample using one GPU. While the complexity is similar to the ECCT

Table 1: A comparison of our method to the baselines on codes seen during pretraining. Higher is better. BP and learned BP results are provided for either 5 (50) iterations in the first (second) line.

| Method | BP | | | Hyp BP | | | ARBP | | | ECCT | | | Ours | | |
|---|---|---|---|---|---|---|---|---|---|---|---|---|---|---|---|
| $E_b/N_0$ | 4 | 5 | 6 | 4 | 5 | 6 | 4 | 5 | 6 | 4 | 5 | 6 | 4 | 5 | 6 |
| BCH(63,36) | 3.72 | 4.65 | 5.66 | 3.96 | 5.35 | 7.20 | 4.33 | 5.94 | 8.21 | 4.56 | 6.37 | 8.85 | 4.53 | 6.38 | 9.10 |
| | 4.03 | 5.42 | 7.26 | 4.29 | 5.91 | 8.01 | 4.57 | 6.39 | 8.92 | | | | | | |
| BCH(127,120) | NA | NA | NA | NA | NA | NA | NA | NA | NA | 4.70 | 6.37 | 8.95 | 4.62 | 6.33 | 8.95 |
| Reed Solomon(21,15) | NA | NA | NA | NA | NA | NA | NA | NA | NA | 5.71 | 7.42 | 9.11 | 5.71 | 7.28 | 9.12 |
| Reed Solomon(60,52) | NA | NA | NA | NA | NA | NA | NA | NA | NA | 5.53 | 7.54 | 9.98 | 5.47 | 7.59 | 10.21 |
| POLAR(32,16) | NA | NA | NA | NA | NA | NA | NA | NA | NA | 6.57 | 8.94 | 11.91 | 6.36 | 8.36 | 11.49 |
| POLAR(64,48) | 3.52 | 4.04 | 4.48 | 4.25 | 5.49 | 7.02 | 4.77 | 6.30 | 8.19 | 6.21 | 8.32 | 10.71 | 6.06 | 8.21 | 10.90 |
| | 4.26 | 5.38 | 6.50 | 4.59 | 6.10 | 7.69 | 5.57 | 7.43 | 9.82 | | | | | | |

$\big(\mathcal{O}(N(d^2(2n-k)+n^2d))\big)$, the number of parameters of the FECCT is much smaller, since it is independent of the code's length, contrary to the ECCT, which requires $\mathcal{O}((2n-k)d+n(2n-k))$ more parameters for the input and output encodings.

## 5 EXPERIMENTS

To evaluate our method, we train one model of the proposed architecture with four classes of linear codes: Low-Density Parity Check (LDPC) codes (Gallager, 1962), Polar codes (Arikan, 2008), Reed Solomon codes (Reed & Solomon, 1960) and Bose–Chaudhuri–Hocquenghem (BCH) codes (Bose & Ray-Chaudhuri, 1960). All parity check matrices are taken from Helmling et al. (2019).

We compare our method with the BP algorithm (Pearl, 1988), the recent Autoregressive hyper-network BP of (Nachmani & Wolf, 2021) (AR BP), and the SOTA ECCT (Choukroun & Wolf, 2022a) with the same number of layers. Note that LDPC codes are specifically designed for BP-based decoding (Richardson et al., 2001).

The results are reported as negative natural logarithm bit error rates (BER) for three different normalized SNR values ($Eb/N_0$), following the conventional testing benchmark, e.g., Nachmani & Wolf (2019); Choukroun & Wolf (2022a). BP-based results are obtained after $L = 5$ BP iterations in the first row (i.e. 10-layer neural network) and *at convergence* results in the second row are obtained after $L = 50$ BP iterations (i.e., 100-layer neural network). During testing, at least $10^5$ random codewords are decoded, to obtain at least $50$ frames with errors at each SNR value. We trained and tested all reported ECCT results to ensure that the models were trained on the same parity-check matrices. All the other baseline results were obtained from the corresponding papers, omitting the codes that have not been tested by these baselines (NA). We refer the reader to Choukroun & Wolf (2022a;b) for complexity and performance comparisons with specialized decoders such as BP and SCL Tal & Vardy (2015).

In Table 1 we present the performance of our model on codes seen during training. As can be seen, our method can outperform the state of the art or remain very close to it.

Table 2 depicts the zero-shot performance of our model, i.e., the performance on unseen codes. Evidently, our method outperforms the state of the art even on some of the unseen codes. However, for codes from families that are underrepresented in the training set, such as Polar codes, a per-code transformer may outperform our zero-shot performance. Reassuringly, the results for BCH(255,163) show that our model is able to generalize to an unseen length that exceeds the length of the longest training code by a factor of two. We also provide zero-shot generalization performance on the BCH(1023,1013) code (seven times longer than the larger code in the training set) in Appendix H.

Finally, in Figure 4, we present the performance improvement when we fine-tune a given code, i.e., train on the code of interest for 500 epochs. Other fine-tuning strategies can also be more beneficial. As can be seen, the proposed fine-tuning enables us to further outperform or close the gap with the state of the art, where needed. We provide BER curves on several codes in Appendix F.

Table 2: A comparison of our method with literature baselines on zero-shot learning, i.e., the performance of the FECCT is provided on new codes only. Higher is better. BP and learned BP results are provided for either 5 (50) iterations in the first (second) line.

| Supervision | Unlearned | | | Fully supervised | | | | | | | | | Zero-Shot | | |
|---|---|---|---|---|---|---|---|---|---|---|---|---|---|---|---|
| Method | BP | | | Hyp BP | | | ARBP | | | ECCT | | | Ours | | |
| $E_b/N_0$ | 4 | 5 | 6 | 4 | 5 | 6 | 4 | 5 | 6 | 4 | 5 | 6 | 4 | 5 | 6 |
| BCH(63,45) | 4.08 | 4.96 | 6.07 | 4.48 | 6.07 | 8.45 | 4.80 | 6.43 | 8.69 | 5.18 | 7.24 | 10.20 | 5.18 | 7.32 | 10.31 |
| | 4.36 | 5.55 | 7.26 | 4.64 | 6.27 | 8.51 | 4.97 | 6.90 | 9.41 | | | | | | |
| BCH(63,51) | 4.34 | 5.29 | 6.35 | 4.64 | 6.08 | 8.16 | 4.95 | 6.69 | 9.18 | 5.63 | 7.96 | 11.22 | 5.71 | 8.07 | 11.31 |
| | 4.50 | 5.82 | 7.42 | 4.80 | 6.44 | 8.58 | 5.17 | 7.16 | 9.53 | | | | | | |
| BCH(127,92) | NA | NA | NA | NA | NA | NA | NA | NA | NA | 4.10 | 5.71 | 8.38 | 4.11 | 5.84 | 8.79 |
| BCH(255,163) | NA | NA | NA | NA | NA | NA | NA | NA | NA | 3.34 | 4.13 | 5.80 | 3.34 | 4.13 | 5.76 |
| CCSDS(128,64) | 6.55 | 9.65 | 13.78 | 6.99 | 10.57 | 15.27 | 7.25 | 10.99 | 16.36 | 6.77 | 10.51 | 15.90 | 6.52 | 9.67 | 15.01 |
| | - | - | - | - | - | - | - | - | - | | | | | | |
| CCSDS(32,16) | NA | NA | NA | NA | NA | NA | NA | NA | NA | 5.93 | 7.77 | 10.02 | 5.23 | 7.00 | 9.21 |
| POLAR(128,86) | 3.80 | 4.19 | 4.62 | 4.57 | 6.18 | 8.27 | 4.81 | 6.57 | 9.04 | 6.39 | 9.08 | 12.70 | 5.53 | 7.90 | 11.29 |
| | 4.49 | 5.65 | 6.97 | 4.95 | 6.84 | 9.28 | 5.39 | 7.37 | 10.13 | | | | | | |
| POLAR(64,32) | 3.52 | 4.04 | 4.48 | 4.25 | 5.49 | 7.02 | 4.77 | 6.30 | 8.19 | 6.91 | 9.18 | 12.34 | 5.88 | 7.91 | 10.76 |
| | 4.26 | 5.38 | 6.50 | 4.59 | 6.10 | 7.69 | 5.57 | 7.43 | 9.82 | | | | | | |

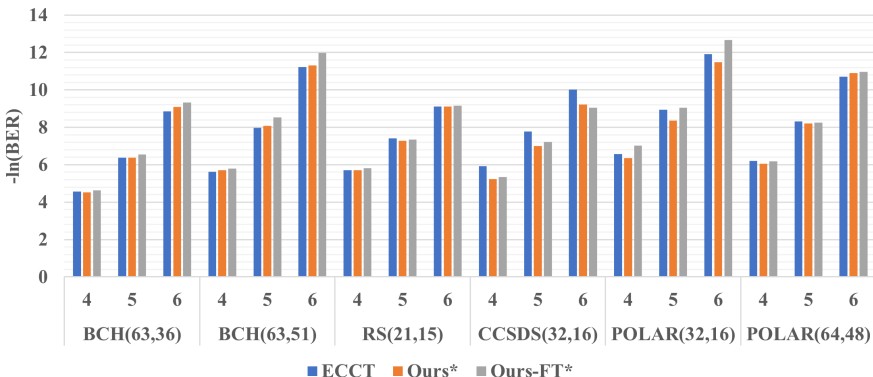

Figure 4: A comparison of our method with the SOTA ECCT, the pretrained FECCT and the fine-tuned FECCT (Ours-FT) on the code of interest. Higher is better.

## 5.1 ABLATION STUDY

**Architectural contributions:** we present in Table 3 the impact of the multiple architectural propositions on performance. Besides the method's invariance to code and length, these results demonstrate the beneficial impact of each of the contributions on the obtained accuracy. Most crucially, without the suggested distanced-based masking (using ECCT's mask on bits for distances smaller than 3), the performance drops when both the input and output are invariant, since there is no position encoding. Interestingly, the invariant initial embedding inductive bias provides better accuracy while having less parameters. Finally, we provide in Figure 5 the illustration of the fine-grained learned mapping of the graph distance. It is interesting to observe that the model seems to assign the most impactful mapping for the elements distanced by one and two nodes, remarkably matching the ECCT's two-ring heuristic. Illustrations of typical self-attention maps are given in Appendix G.

**Generalization contribution:** We provide a generalization analysis of the framework in Appendix E where we show the advantage of the proposed training strategy for generalization.

## 6 DISCUSSION AND LIMITATIONS

The proposed framework permits the efficient integration of a single code-invariant neural decoder into base stations and error correction system on chips, correcting multiple code families at different rates. This can potentially save die space as well as storage and memory usage while matching or even outperforming the state-of-the-art neural decoders. Also, the development of universal neural

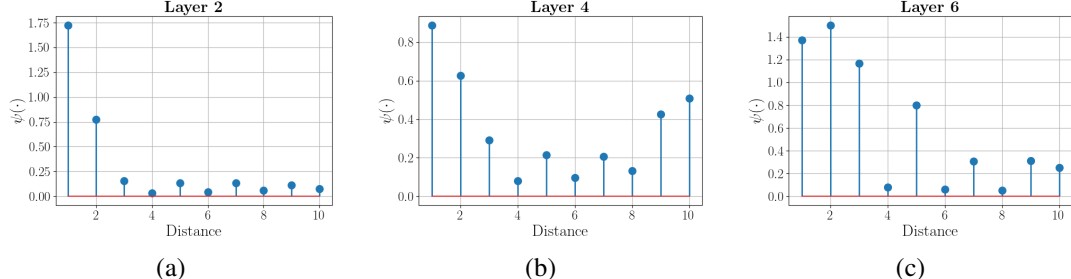

(a)                                    (b)                                    (c)

Figure 5: Absolute values of the learned mapping of the proposed FECCT with respect to distance for three layers of the model among the six. The mapping of all the layers is given in Appendix D.

Table 3: A comparison of the impact of each one of the presented contributions on the performance. All the models have a $N = 2, d = 32$ architecture and follow the same ECCT's training setting of 1000 epochs. *DM* refers to the proposed Distanced-based Masking strategy, *II* refers to the proposed code Invariant Initial embedding, and *IO* refers to the suggested Invariant Output parity-check aware aggregation scheme. We note the ECCT has 15% more parameters than the FECCT.

| Method | POLAR(64,32) | | | BCH(63,45) | | |
|---|---|---|---|---|---|---|
| | 4 | 5 | 6 | 4 | 5 | 6 |
| **ECCT** | 4.12 | 5.22 | 6.67 | 4.45 | 5.81 | 7.65 |
| ECCT + **II** | 4.27 | 5.54 | 7.14 | 4.52 | 5.98 | 7.92 |
| ECCT + **IO** | 4.44 | 5.73 | 7.40 | 4.41 | 5.76 | 7.62 |
| ECCT + **II** + **IO** | 4.09 | 5.26 | 6.80 | 4.31 | 5.62 | 7.41 |
| ECCT + **DM** | 4.44 | 5.73 | 7.37 | 4.74 | 6.34 | 8.53 |
| ECCT + DM + **II** | 4.44 | 5.73 | 7.37 | 5.17 | 7.07 | 9.59 |
| ECCT + DM + **IO** | 4.36 | 5.64 | 7.32 | 4.53 | 6.01 | 8.03 |
| **FECCT**: ECCT + DM + II + IO | 4.36 | 5.64 | 7.32 | 4.52 | 5.98 | 8.05 |

decoders allows for the differentiable optimization of codes. For example, existing or new codes can be improved iteratively to enable more effective decoding in higher SNRs.

It is important to note that the proposed universal *decoding architecture* can be improved further by applying it together with other neural *decoding algorithms*, such as the denoising diffusion ECC paradigm (Choukroun & Wolf, 2022b), which was evaluated with the ECCT architecture backbone.

As our experiments reveal, dataset diversity is important for generalization and performance. Currently, the framework we trained on codes obtained from the dataset of Helmling et al. (2019) which are highly unbalanced, with an overrepresentation of BCH codes. We also note that the parity check matrices of the Polar codes have been standardized following Choukroun & Wolf (2022a) which, besides the under-representation of Polar codes, may be why the FECCT is less performant than ECCT on zero-shot tasks for these specific codes.

## 7 CONCLUSION

We present a novel foundation model based on the Transformer architecture for the decoding of algebraic block codes. The proposed model allows effective representation of interactions between the code's elements in an invariant fashion, and the processing of any code of any length. The proposed method reaches and even outperforms the state of the art, while having fewer parameters and being code- and size-invariant, enabling the potential efficient deployment of a single universal decoder for ECC. The next steps should focus on the deployment of the method on existing error correction embedded systems via the utilization of Transformer acceleration methods along with extensive tuning and training of the codes and channels of interest. Additionally, the definition of a universal differentiable neural decoder may open the door to the optimization of codes and even to the learning of new families of codes.

## 8 ACKNOWLEDGEMENTS

This project has received funding from the Tel Aviv University Center for AI and Data Science (TAD) and the Blavatnik Computer Science Research Fund. The contribution of the first author is part of a PhD thesis research conducted at Tel Aviv University.

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

## A   ILLUSTRATION OF THE COMMUNICATION SYSTEM

We provide in Figure 6 an illustration of the proposed communication and decoding setting.

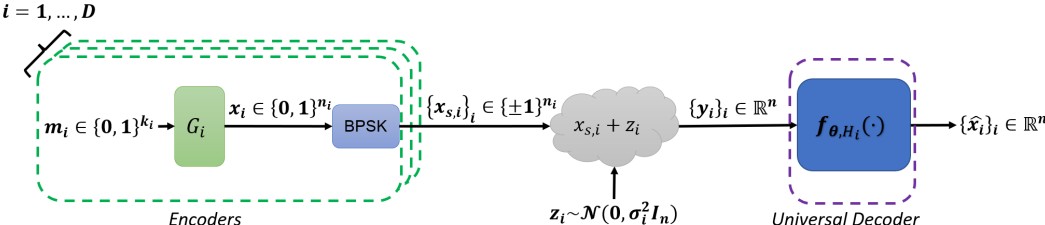

Figure 6: Illustration of the communication system. In the suggested foundation model setting, a *single* decoder $f_\theta$ is proposed, capable of correcting any code at any length, given the code's parity-check matrix. Here, $D$ denotes the number of different samples and noises, indexed by $i$, to be processed by the decoder.

## B   LOSS OF ADJACENCY INFORMATION IN ECCT

To illustrate the loss of connectivity information from the ECCT's mask we provide in Figure 7 a description of the graph reconstruction from the ECCT's mask for the $(3, 1)$ repetition code. As can be seen, the Tanner graph reconstructed from the mask cannot describe graph connectivity accurately because of the hard distance (one and two-ring) thresholding. For this code, assuming no connectivity between the variable nodes prior would allow faithful reconstruction.

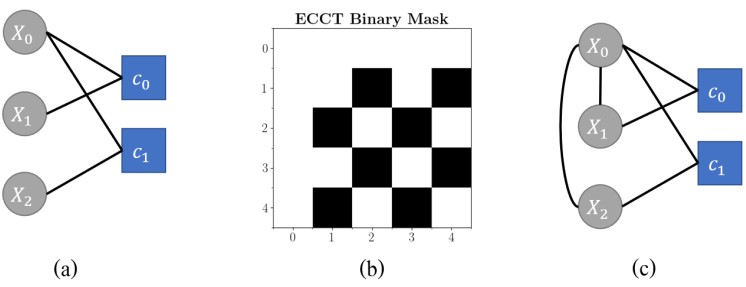

Figure 7: For the $(3, 1)$ repetition code: (a) the original Tanner graph, (b) the induced ECCT's mask, (c) the "Tanner" graph derived from the mask.

## C   DESCRIPTION OF THE CODES

We present in Figure 8 the statistics of the codes used in this work. The codes were randomly sampled for the creation of the datasets.

- TRAIN: BCH_127_106_3_strip.txt
- TRAIN: BCH_127_113_2_strip.txt
- TRAIN: BCH_127_120_1_strip.txt
- TRAIN: BCH_127_64_10_strip.txt
- TRAIN: BCH_127_71_9_strip.txt
- TRAIN: BCH_127_78_7_strip.txt
- TRAIN: BCH_127_85_6_strip.txt
- TRAIN: BCH_127_99_4_strip.txt
- TRAIN: BCH_15_11_1_strip.txt
- TRAIN: BCH_31_11_5_strip.txt
- TRAIN: BCH_31_16_3_strip.txt
- TRAIN: BCH_31_21_2_strip.txt
- TRAIN: BCH_63_30_6_strip.txt
- TRAIN: BCH_63_36_5_strip.txt
- TRAIN: BCH_63_39_4_strip.txt
- TRAIN: BCH_63_51_2_strip.txt
- TRAIN: BCH_7_4_1_strip.txt
- TRAIN: Hopt_BCH_127_64_10_1400ones.alist

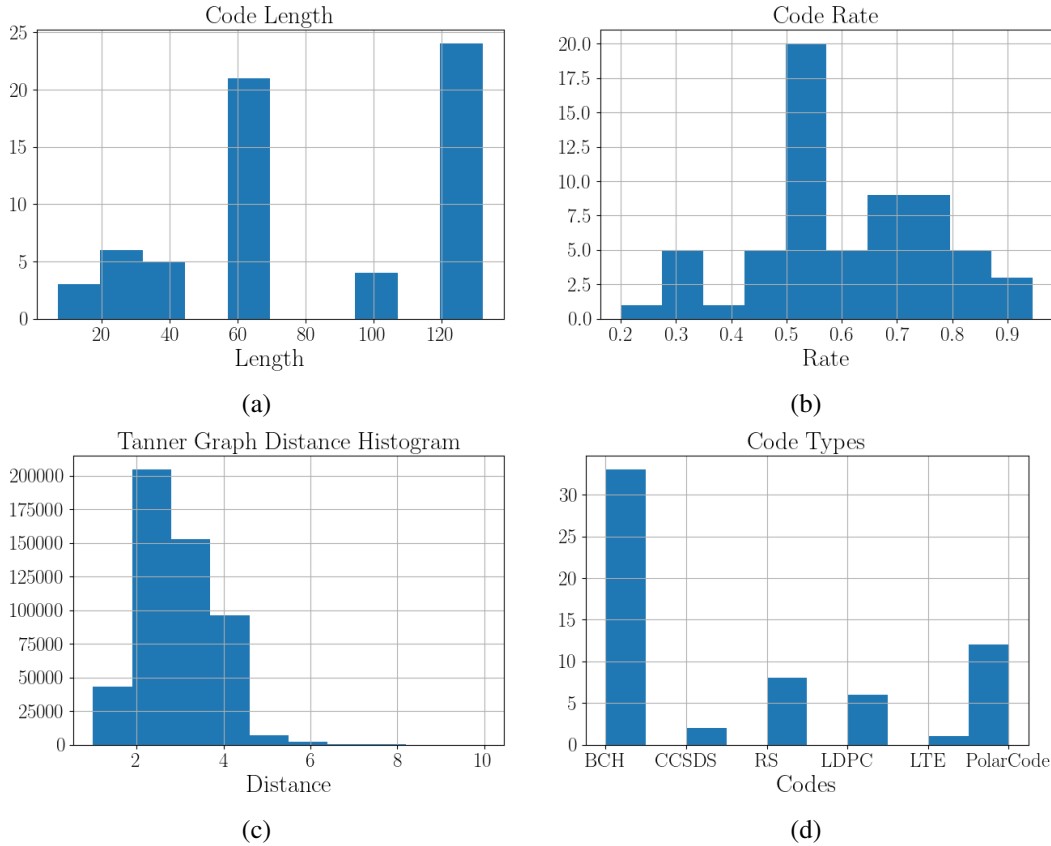

Figure 8: We present statistics regarding the codes used in this work: the codes' lengths (a), the codes' rates (b), the Tanner graph distances (c), and the distribution of the code types.

- TRAIN: Hopt_BCH_127_71_9_1292ones.alist
- TRAIN: Hopt_BCH_127_78_7_1372ones.alist
- TRAIN: Hopt_BCH_127_85_6_1344ones.alist
- TRAIN: Hopt_BCH_127_99_4_1232ones.alist
- TRAIN: Hopt_BCH_63_36_5_384ones.alist
- TRAIN: Hopt_BCH_63_39_4_336ones.alist
- TRAIN: Hopt_BCH_63_45_3_288ones.alist
- TRAIN: Hopt_RS_15_11_2_272ones.alist
- TRAIN: Hopt_RS_15_13_1_192ones.alist
- TRAIN: Hopt_RS_15_3_6_208ones.alist
- TRAIN: Hopt_RS_15_5_5_272ones.alist
- TRAIN: Hopt_RS_15_7_4_288ones.alist
- TRAIN: Hopt_RS_15_9_3_308ones.alist
- TRAIN: Hopt_RS_7_3_2_54ones.alist
- TRAIN: Hopt_RS_7_5_1_48ones.alist
- TRAIN: LDPC_N128_K64_GF256_UNBPB_bi.alist
- TRAIN: LDPC_N128_K64_GF256_bi.alist
- TRAIN: LDPC_N96_K48_GF256_d1_bi.alist
- TRAIN: LDPC_N96_K48_GF64_BI.alist
- TRAIN: LDPC_N96_K48_GF64_bi.txt
- TRAIN: LDPC_N96_K48_P8_set0_dmin10.txt
- TRAIN: PolarCode_N128_K43.txt.bz2
- TRAIN: PolarCode_N128_K96.txt.bz2
- TRAIN: PolarCode_N32_K11.txt.bz2
- TRAIN: PolarCode_N32_K16.txt.bz2
- TRAIN: PolarCode_N32_K22.txt.bz2
- TRAIN: PolarCode_N32_K24.txt.bz2
- TRAIN: PolarCode_N64_K22.txt.bz2
- TRAIN: PolarCode_N64_K32.txt.bz2
- TRAIN: PolarCode_N64_K43.txt.bz2
- TRAIN: PolarCode_N64_K48.txt.bz2
- TEST: BCH_127_92_5_strip.txt
- TEST: BCH_15_7_2_strip.txt
- TEST: BCH_31_26_1_strip.txt
- TEST: BCH_63_45_3_strip.txt
- TEST: BCH_63_57_1_strip.txt
- TEST: CCSDS_ldpc_n128_k64.alist
- TEST: CCSDS_ldpc_n32_k16.alist
- TEST: Hopt_BCH_127_92_5_1148ones.alist
- TEST: Hopt_BCH_63_30_6_396ones.alist
- TEST: Hopt_BCH_63_51_2_288ones.alist
- TEST: PolarCode_N128_K64.txt.bz2
- TEST: PolarCode_N128_K86.txt.bz2
- TEST: eBCH_128_64_strip.txt

# D LEARNED MAPPINGS VISUALIZATIONS

We present in Figure 9 the values of the learned mapping of the Tanner graph distances for all the layers of the proposed FECCT. In all layers, we see a prominence to the mapping of distance 1 or distance 2, with layer 1 being the only case in which these two do not have the highest values. This is reminiscent of the fixed map in ECCT, where the first two values are mapped to one, and the rest to zero.

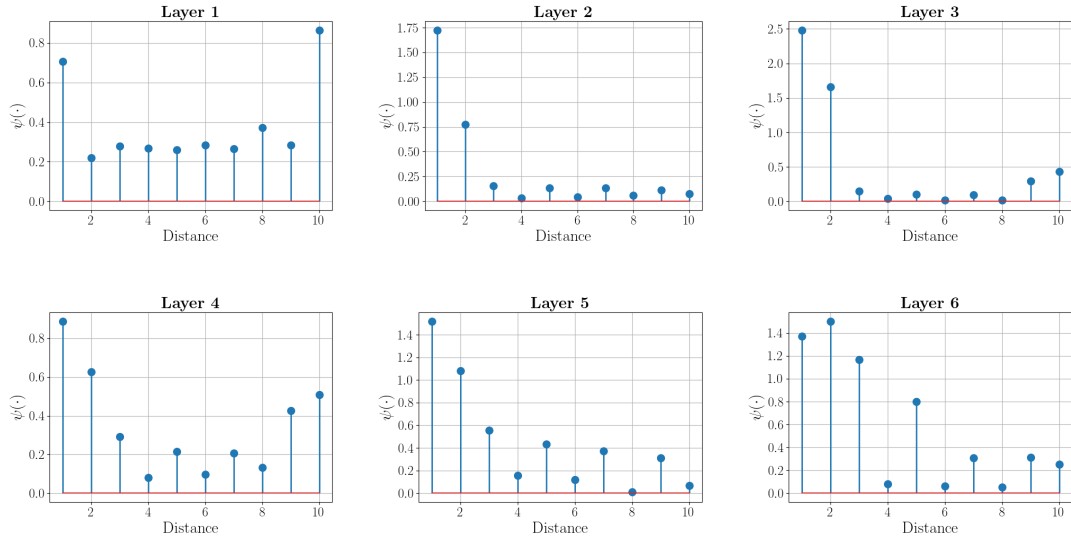

Figure 9: Absolute values of the learned mapping of the proposed FECCT with respect to distance for the six layers of the model.

# E GENERALIZATION ANALYSIS

To illustrate the importance of dataset diversity, we provide in Table 4 a comparison with a FECCT trained on a single code only (POLAR(64,48)). The lack of generalization capacity of the model compared to the proposed training strategy is evident. The performance on the code of interest is only slightly better for the single code version.

Table 4: A comparison between the proposed fully trained FECCT and a FECCT trained on the PO-LAR(64,48) code only (FECCT-single), demonstrating the generalization benefits of the proposed method. Zero-shot codes are marked by a star. Higher is better.

| Method | FECCT - single | | | FECCT | | |
|---|---|---|---|---|---|---|
| | 4 | 5 | 6 | 4 | 5 | 6 |
| POLAR(64,48) | 6.35 | 8.50 | 11.12 | 6.06 | 8.21 | 10.96 |
| POLAR(128,86)* | 3.90 | 5.36 | 7.57 | 5.53 | 7.90 | 11.29 |
| BCH(63,36) | 4.01 | 5.42 | 7.30 | 4.53 | 6.38 | 9.10 |
| BCH(63,51)* | 4.65 | 6.35 | 8.73 | 5.71 | 8.07 | 11.31 |
| Reed Solomon(21,15) | 4.25 | 4.62 | 4.97 | 4.56 | 6.83 | 10.51 |
| Reed Solomon(60,52) | 3.68 | 3.81 | 3.77 | 5.47 | 7.49 | 10.24 |
| CCSDS(128,64)* | 2.90 | 3.42 | 4.30 | 6.52 | 9.67 | 15.01 |
| CCSDS(32,16)* | 4.10 | 4.54 | 4.43 | 5.23 | 7.00 | 9.21 |

## F  BER Curves

We depict in Figure 10 the BER curves of the ECCT, the zero-shot FECCT, and the fine-tuned FECCT.

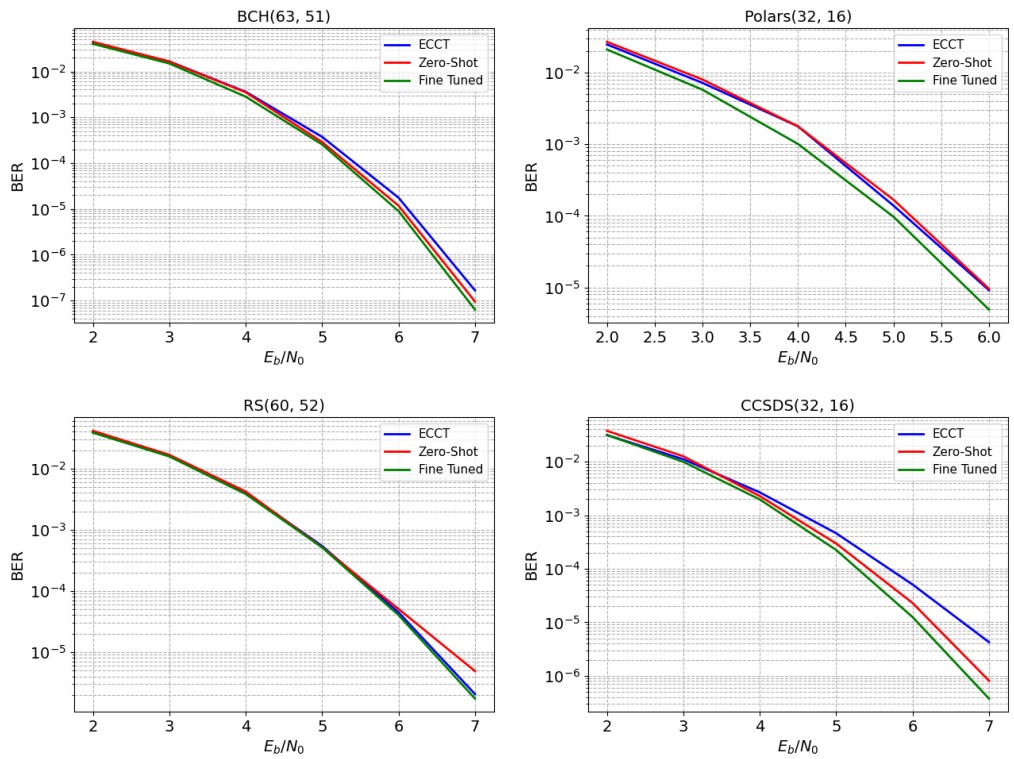

Figure 10: BER curves of the ECCT, the zero-shot FECCT, and the fine-tuned FECCT for four different codes.

## G  Self Attention Maps

We depict in Figure 12 the self-attention maps at the different layers of the model for different $E_b/N_0$ values averaged over a 2048 sample batch. At each layer, we present the original self-attention map as well as the distance-filtered one as described in Eq (7). In Figure 11, we present the sefl-attention maps for a given sample. We can observe that the attention is mostly focused on the syndrome values in order to perform the decoding, while in the final layer, the focus is also transferred to the information bits.

## H  Experiment on large code

We provide in Figure 13 the zero-shot performance of the pretrained FECCT on the BCH(1023,1013) code. This code is seven times the size of the largest code in our training set, and 13 times the mean code length in the training set. The baseline is taken from Helmling & Scholl (2016). Evidently, the zero-shot decoding is approximately one dB worse than maximum-likelihood decoding and better than the Hard Decision Decoding baseline.

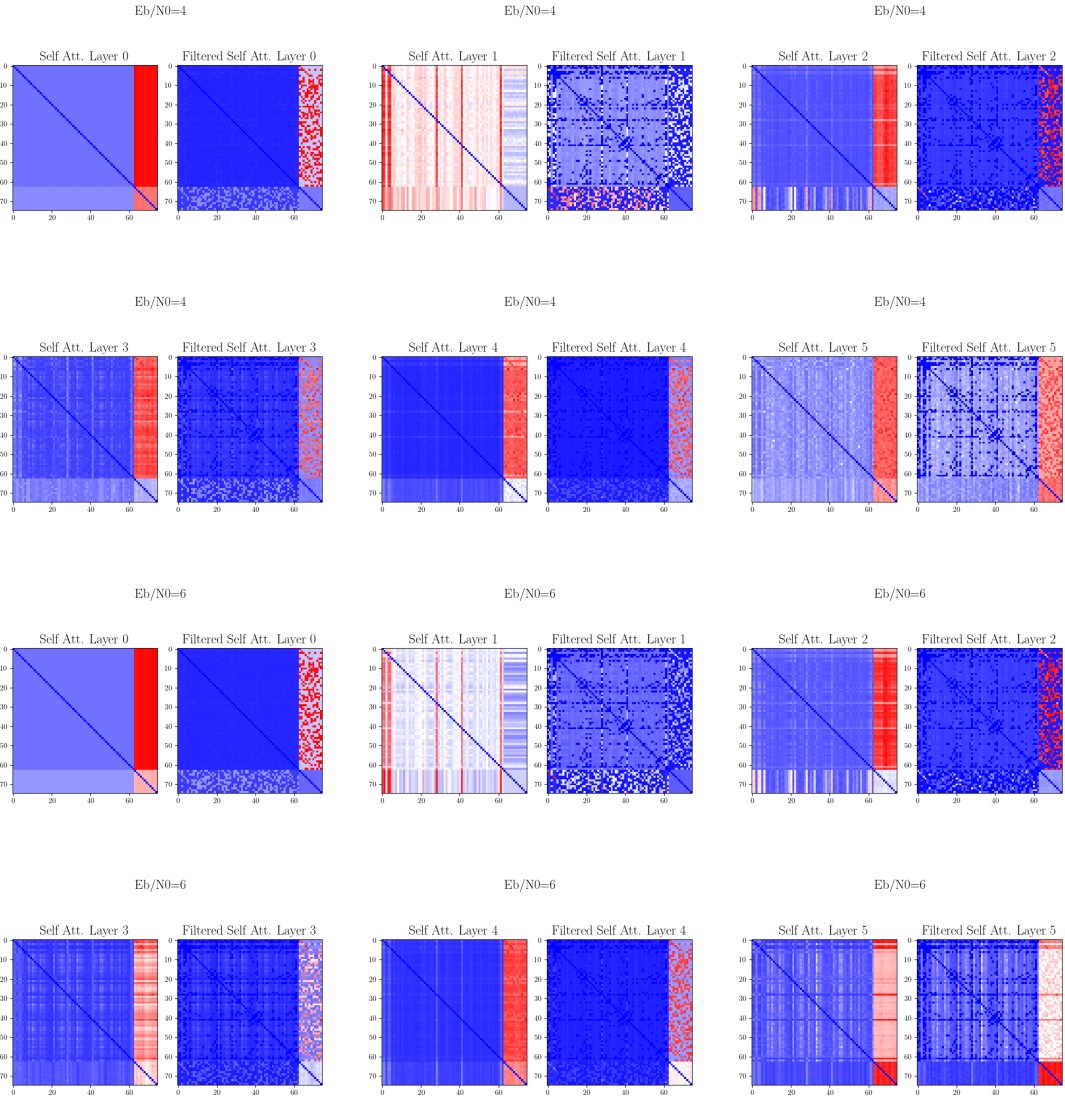

Figure 11: Absolute values of the self-attention maps at the six different layers of the model for two different $E_b/N_0$ values for the BCH(63,51) code. At each layer, we provide the self-attention before and after the Tanner graph distance filtering. The self-attention tensor is averaged over the batch and head dimensions.

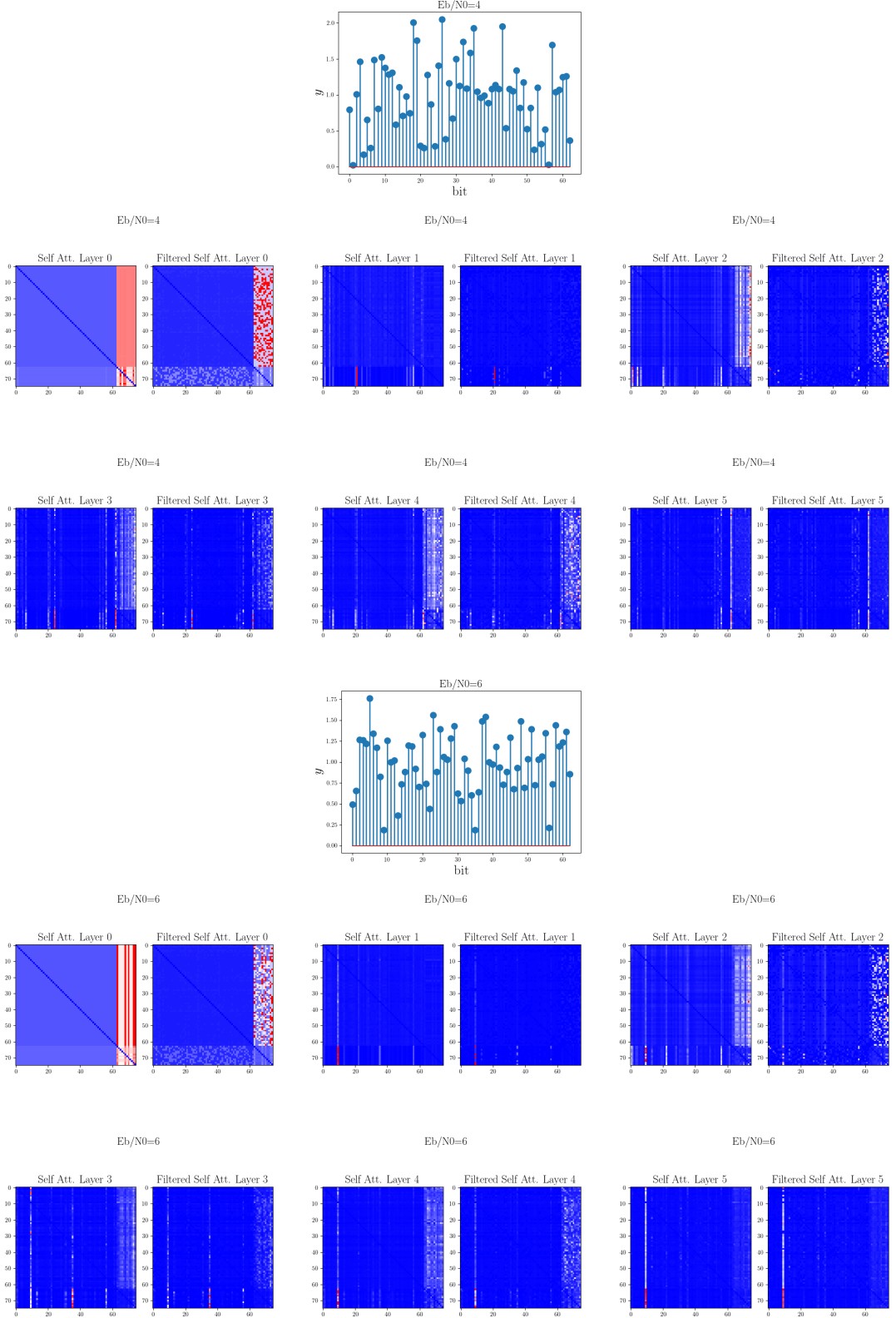

Figure 12: Absolute values of the channel output values $y$ followed by the corresponding absolute values of the self-attention maps at the six different layers of the model for two different $E_b/N_0$ values for the BCH(63,51) code. At each layer, we provide the self-attention before and after the Tanner graph distance filtering. The self-attention tensor is averaged over the head dimension.

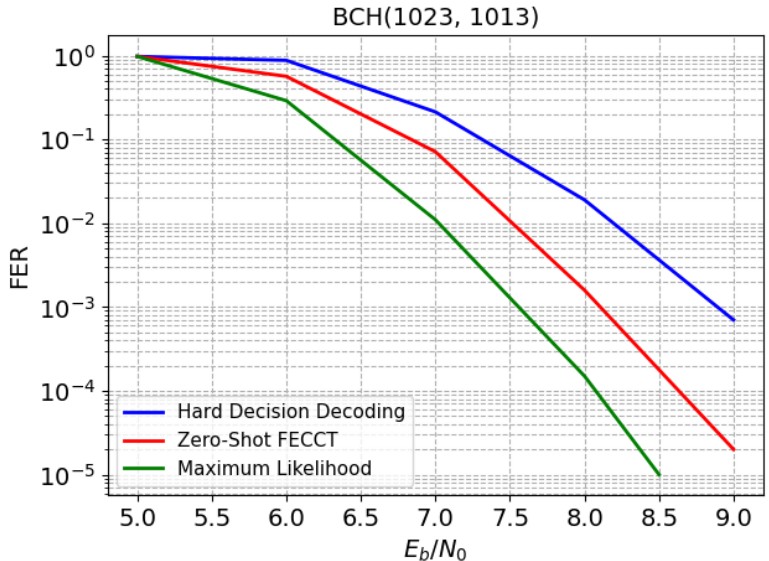

Figure 13: Zero-shot FER performance of the pretrained FECCT, Hard Decision Decoder and Maximum-likelihood decoders on the BCH(1023,1013) code

