# OpenReview forum: "A Foundation Model for Error Correction Codes"
_ICLR.cc/2024/Conference — ICLR 2024 poster_

### Official Review · Reviewer_jdxb · 2023-10-30

**Soundness:** 4 excellent
**Presentation:** 3 good
**Contribution:** 3 good
**Rating:** 8
**Confidence:** 4

**Summary:**

The paper builds on top of Error Correction Code Transformer(ECCT), made a few structural change to build a "foundational" ECCT (FECCT) model for block code decoding.

FECCT is a generalized version of ECCT:

(1) FECCT is not length-dependent, H-dependent, and not even code-dependent, thus can be trained once can be used for wide range of block codes, with good performance (matching/beating ECCT mostly, and beat BP by a large margin.). Some finetuning can even lead to better performance.

(2) FECCT could an important milestone of deploying neural decoders to real world given its potential, still a lot of hard work is required till that day.

**Strengths:**

(1) Overall, FECCT is very plausible method, since FECCT is more like an "decoding algorithm" rather than simply "neural decoder". My definition of "decoding algorithm" means the input is H matrix  and received codeword  (just like BP algorithm), and the output should be decoded message. While other neural decoder has dependency on code/length/H/etc. As of my understanding, FECCT has learned some interesting advanced BP-like algorithm, that can be beneficial for a wide family of block codes.

(2) FECCT is built on top of ECCT, the generalization performance on non-zero codewords are preserved, which makes training feasible. FECCT's H-dependent attention is a generalized version of ECCT's attention, which lead to better decoding capability. The proposed neural structure makes sense, and lead to good generalization performance.

(3) The experiment on unseen code with different code family and block length are interesting, which make (1)'s claim stronger that FECCT is more of an "decoding algorithm".

Overall, this becomes an interesting work, at least for neural decoder research, first time shows that a "decoding algorithm" rather than a complicated mapping can be learned.

**Weaknesses:**

1. The experiments are mostly built on short block codes (<128, test unseen for 255 at most), while typical capacity-approaching codes such as QC-LDPC has much longer block length. Performance on long block length is going to make this paper stronger, due to long block code's capacity-approaching performance.

2. Interpretability: FECCT should have been learned some interesting algorithm, that can be interpreted as an advanced version of BP. We do see some part of interpretations in the appendix, but not solid enough to get insight on what FECCT's algorithm means.

3. Complexity of network: attention-based neural network are very complex. Deploying the FEECT to any real world production requires some hard work on complexity reduction. In its current form, I am not seeing FECCT can be deployed to modem in short time.  Note that channel coding are heavily used in all modern wireless communication systems, which requires minimal latency, high throughput, and low cost.

**Questions:**

N/A

---

> ### Author Response · Authors · 2023-11-14
>
> We thank the reviewer for the very supportive review.
>
> ## Code Length
> As with most neural decoding works (cf. Related Works section), we focus on short-length codes for three main reasons.
>
> (i) On very large codes, some decoders (e.g., BP, SC) have been proven to achieve performance close to ML decoding on their respective codes of interest (e.g., LDPC, Polar, respectively), rendering the potential advantage of neural decoding solutions nonexistent/irrelevant.
>
> (ii) The training and/or deployment of the existing neural decoding techniques is not trivial/straightforward for very large codes (i.e., code lengths of thousands of bits).
>
> (iii)  The emergence of applications driven by the Internet of Things created the requirement for optimal decoders of short to moderate codes. For example, 5G Polar codes have code lengths of 32 to 1024 for uplink (UCI) or 128 to 512 for downlink [1].
> These points are emphasized in the revision.
>
> [1]  ETSI 3GPP TS 38.212 “5G NR Multiplexing and channel coding”, v.16.5.0, 2021-03.
>
> ## Interpretability
> We provide in the revision’s Appendix G the depiction of typical learned self-attention matrices.  As can be observed, initially the attention is mostly focused on the syndrome values in order to perform the decoding, while in the final layer, the focus is also transferred to the information bits.
>
> ## Integration
> While being out of the scope of the paper, the deployment of the FECCT is indeed a challenge, requiring the application of many deep-learning acceleration techniques such as quantization (binarization) and sparsification (mainly of the self-attention module) as well as its FPGA/ASIC integration.

---

### Official Review · Reviewer_w8i1 · 2023-10-31

**Soundness:** 3 good
**Presentation:** 3 good
**Contribution:** 2 fair
**Rating:** 3
**Confidence:** 4

**Summary:**

The paper attempts to develop foundation model for Error Correction Codes that is trained on large data so that it can be used later for any downstream task. Specifically, authors aim to adapt the Transformer input embedding for robustness to code length variations. To learn the code structure, they use the positional embedding, that is integrated into the self-attention via a learned mapping of the node distances in the Tanner graph. Moreover, for code awareness and channel noise prediction, the paper employs a size-invariant prediction module that is conditioned on the parity-check matrix. In simulations, they tested on codes that are unseen during training. They showed that the proposed FECCT method matches or sometimes perform better than the
state of art.

**Strengths:**

-the paper takes a foundational approach to the decoding problem in error correcting codes, which is intellectually interesting. Clearly being able to decode any type of code is an interesting intriguing  exercise.
- the paper advanced the design of generalist decoders relative to existing generalist decoders by using new embedding and grounding techniques.

**Weaknesses:**

-Although the design of foundational decoders are very interesting intellectual exercise, the real world impact of it is close to none if not zero.  The reason is that error correcting codes are designed and deployed once and their training is not a big deal even if someone takes deep neural network decoders as opposed to classical BP methods. But more importantly, there is another argument against the value of these generalist decoders: The important thing to recall is that capacity achieving codes exists for long codes and their BP decoders are close to ML performance, I.e., optimal decoders. So there is no gain of these deep neural decoders in long codes. The focus should be short codes for which we do not have good BP decoders. However, for short codes, we really do not need foundational decoders as one can design and easily train specialized neural decoders for the short codes that will very likely beat the performance of generalist decoders for all lengths. It is clear to believe that a generalist decoder will not be able to perform a specialized deep neural decoder for short lengths, unless the authors can show their generalist decoder can beat the performance of state of art short length (less that 200 bits) code decoders, specialized for that specific code length.

-what is the performance in the tables? I see that 3 different values of Eb/N0 is used as the channel input signal (bit) power to noise ratio but What about reported numbers as performance. What are they? I like to see how these reported numbers translate to the error rate performance, as it is the only thing that matters in communication. It does not look like that the authors picked a particular error rate and report the corresponding Required Eb/N0 to achieve such an error rate. Because in that case the lower number is associated with the better scheme not the higher (as the authors stated in the paper).

-the authors need to compare their scheme with Choukroun & Wolf (2022b) which is shown to be superior to ECCT.

**Questions:**

please compare the proposed generalist decoder at short lengths (less than 200 bits) with that of specialized decoders at those lengths. Because as I pointed out this is where these foundational decoders would show value if any.

-please plot error rate plots rather than the reporting used in the paper which is not insightful.

-it would be helpful to compare your proposed work with that of
Choukroun & Wolf (2022b) which extends and enhances ECCT via Denoising diffusion in error correction codes and have far superior performance than ECCT.

-the paper novelty is arguable in light of ECCT design architecture. For the most part following similar development as ECCT. Can the authors elaborate on the novelty relative to ECCT.

**Details Of Ethics Concerns:**

There is none.

---

> ### Author Response · Authors · 2023-11-14
>
> We thank the reviewer for the valuable feedback. We believe several points have been misunderstood and we have modified the manuscript to make it clearer.
>
> >“The reason is that error correcting codes are designed and deployed once and their training is not a big deal even if someone takes deep neural network decoders as opposed to classical BP methods”
>
> While we agree with the reviewer that training is not a major limitation compared to deployment, we respectfully disagree with the reviewer regarding the integration/deployment side. Current error-correction systems (e.g., base stations) can/need to decode codes of different lengths and/or different families. For example, LDPC codes of different lengths are decoded on HW by omitting some of the integrated processing engines of the BP implementation for shorter codewords.
>
> Moreover, deploying one single neural decoder for several types of codes and/or lengths is fundamentally different from integrating several neural decoders, especially in terms of die space and storage/memory (one model’s weights vs many models’ weights).
>
> We remind here that existing neural decoders are designed and trained on one single code to be integrated into dedicated HW.
>
> This point is emphasized in the revision.
>
> >“However, for short codes, we really do not need foundational decoders as one can design and easily train specialized neural decoders for the short codes that will very likely beat the performance of generalist decoders for all lengths.”
>
> Following our previous answer, the deployment of a specialized neural decoder has major integration drawbacks.
>
>  Most importantly, the main goal of our work was to show our method, while being code/length-invariant (referred to as “generalist decoder” by the reviewer), can match and even outperform the “specialized” state-of-the-art neural decoder.
>
> In addition to the points above that address the direct utility, foundation models support downstream tasks. Specifically, the ability to move quickly between codes enables the sampling and evaluation of random codes. Moreover, since our method defines a universal differentiable neural decoder, it opens the door to the optimization and definition of new families of codes.
>
> These points are emphasized in the revision.
>
>
> >“What is the performance in the tables?What about reported numbers as performance. What are they? I like to see how these reported numbers translate to the error rate performance”
>
> As described in Section 5, third paragraph, the reported numbers are the “negative natural logarithm bit error rates (BER) for three different normalized SNR values (Eb/N0)”, meaning we provide $-\ln(BER)$ (such that higher is better) for three Eb/N0 values. This dense presentation for multiple codes of the BER on the SNR of most interest has been borrowed from many of the previous neural decoding works.
>
> > “please compare the proposed generalist decoder at short lengths (less than 200 bits) with that of specialized decoders at those lengths. Because as I pointed out this is where these foundational decoders would show value if any.”
>
> While our work focuses on the development of neural decoders, the previous work cited by the reviewer (e.g., [1],[2])  already provided a comprehensive comparison with specialized decoders. In our work, we compare with it.
>
> The revised text now points to these results.
>
> >“it would be helpful to compare your proposed work with that of Choukroun & Wolf (2022b) which extends and enhances ECCT via Denoising diffusion in error correction codes and have far superior performance than ECCT.”
>
> We wish to remind the main difference between ECCT and DDECC, as described in Section 6 paragraph 2. While ECCT focuses on the design of a new neural *decoding architecture*, DDECC is a generic iterative *decoding algorithm* applied to a given backbone architecture. As described in [2] Section 4.5, the ECCT has been chosen since it is the state-of-the-art decoding architecture.
>
> Our model focuses on the first family of work (i.e., decoding architecture) that can, same as ECCT, be coupled with DDECC and makes it code and length invariant too. While these results would be interesting to see, they require many more experiments and much heavier resources.
>
> This point is emphasized in the revision.

---

> ### Author Response · Authors · 2023-11-14
>
> >“Can the authors elaborate on the novelty relative to ECCT.”
>
> We believe the paper differs from ECCT as much as ECCT differs from the original Transformer architecture.
>
> ECCT brings three modifications to the original Transformer architecture. (i) An initial bit-wise embedding (i.e., $n$ encoding vectors, with $n$ the code length). (ii) Thresholded binary masking. (iii) A Linear (FC) prediction layer.
>
> FECCT redefined these three contributions in order to (1) make them code/length-invariant, and (2) improve over them (c.f. Table 3).
>
> (i) The initial embedding is restricted to three embedding vectors, one vector for each bit to be modulated by the channel’s output magnitude and a one-hot encoding defined by the syndrome values.
>
> (ii) The masking is totally different in its definition and implementation and is defined as
> $softmax(QK^{T})\odot \psi(\mathcal{G}(H))$ compared to $softmax(QK^{T}+g(H))$.
> Here $g(H)\in \\{-\infty,0\\}$ denotes ECCT’s binary thresholding, while FECCT provides a fine-grained masking based on the Tanner graph distances.
>
> (iii) The prediction module is fundamentally different than a Linear layer and is based on the parity-check matrix in order to remain invariant while being code-aware.
>
> >“-please plot error rate plots rather than the reporting used in the paper which is not insightful.”
>
> We now also provide BER curves for several codes in Appendix F.
>
> ## References
> [1] Error Correction Code Transformer, Y. Choukroun and L. Wolf, NeurIPS 2022
>
> [2] Denoising Diffusion Error Correction Codes, Y. Choukroun and L. Wolf, ICLR 2023

---

> ### Comment · Reviewer_w8i1 · 2023-11-16
> **Short length vs long length & performance issue**
>
> Thanks authors for clarifying a few things. However, I beLieve you misunderstood my inputs. (1) I am questioning the value of this work in real world. Again, I do not debate its intellectual exercise, which is good in my mind. But I disagree with authors in their argument for its real world implications for the following reasons: for long codes, we already have very good capacity achieving codes (e.g., LDPC and polar) that are rate compatible so we don’t have to worry about their rate variations. The authors are likely aware that LDPS codes, for example, can be punctured randomly and create very good rate compatible codes with BP decoding near their ML performance. So we should all agree that no neural based codes are needed for such lengths. Te need is in short lengths from practical impact, that is where all the potential resides for neural solutions. I would be, however, impressed if the authors can find decoders that are good for very short lengths (specially for Reed Muller codes) as we do not have good decoders for such cases.  But the issue is that the authors show in appendix F that Fine tuned ECCT (and even initial ECCT, for the most part) always perform better than their codes in short lengths. So how we can justify the need for generalist decoder from practical point of view?
>
> Regarding hardware complexity of decoding various codes, if the environment is a private network, it is setup to operate on the rate variation of the same type of code so I am not sure we need generalist decoder for that. If it is wireless cellular network, each network provider has its own parameter settings and uses the same code, that are preset, so I do not see any complications arise for decoding.
> So from practical point of view, I would rather to see a work that  contributes to take one rate adaptive code with very good decoder at various rates rather than a generalist decoder for various types of codes and lengths  at these short lengths. Again, I acknowledge that the work is intellectually good but I am not convinced about the usefulness of this line of work.

---

> ### Author Response · Authors · 2023-11-16
>
> Thank you for the prompt reply! We are happy to hear that we have been able to clarify some issues and are very encouraged by the reviewer’s acknowledgment of the value of the work as basic research with good intellectual value.
>
> >The need is in short lengths from practical impact, that is where all the potential resides for neural solutions.
>
> We agree with the reviewer that short codes are (*including our work*) and should be the focus of neural decoders. In the revision, we have added a paragraph to the Related Work section discussing this point.
>
> >But the issue is that the authors show in appendix F that finetuned ECCT (and even initial ECCT, for the most part) always perform better than their codes in short lengths. So how we can justify the need for a generalist decoder from the practical point of view?
>
> This is a misunderstanding that stems from an unfortunate typo we had in the figure captions (now fixed in red). We deeply apologize for this. There is no such thing as zero-shot or finetuned ECCT, since ECCT is trained for a single code. Appendix F, as well as the Tables in the main text, show the performance of the SOTA ECCT compared with the performance of the proposed *Foundational* ECCT (FECCT) on zero-shot (i.e., unseen codes) and finetuned codes (i.e., short training on a code of interest) tasks. The results, along with the ablation study, support the fact that our method, while being length/code/rate-invariant and having fewer parameters, is able to reach and even outperform the code-dedicated SOTA ECCT.
>
> To be clear, for 100% of the cases, our finetuned version outperforms the ECCT. In a limited number of **zero-shot** runs, this is not the case. As described in Section 6, the discrepancy in performance between the different families of codes is due to the underrepresentation of some code families in the training set.
>
> >Each network provider has its own parameter settings and uses the same code, that are preset, so I do not see any complications arise for decoding.
>
> We respectfully disagree. A network provider does not define its own parameters but must acquire/use telecommunication infrastructures and hardware that follow a given standard. For example, 5G supports both LDPC and Polar codes [1], meaning that modern systems (supporting 2G to 5G standards) must be able to decode (on device) different families of codes (e.g., BCH, LDPC, Polar). We can also find similar multiple decoders in [Concatenated error correction code](https://en.wikipedia.org/wiki/Concatenated_error_correction_code).
>
> [1] ETSI 3GPP TS 38.212 “5G NR Multiplexing and channel coding”, v.16.5.0, 2021-03.
>
> >...”it is setup to operate on the rate variation of the same type of code”... “So from practical point of view, I would rather to see a work  that contributes to take one rate adaptive code with very good decoder at various rates rather than a generalist decoder for various types of codes and lengths at these short lengths.”
>
> Existing neural decoders are not rate-adaptive at all. For example, the state-of-the-art ECCT requires a new *distinct* model for each rate (or even length) of interest. Our neural decoder is absolutely rate-adaptive by construction. As far as we can ascertain, this is the first time a neural decoder is length/rate invariant, enabling the potential practical integration of one neural decoder for any rate/length.
>
> If needed, further specialization of the foundation model can be obtained simply by training it on a single family of codes. Since this is against the basic scientific ambition to generalize, we did not attempt this, but it is a valid future direction.
>
> To summarize: our method is both conceptually and architecturally novel. It has multiple advantages on short/moderate codes: (1) The method reaches and outperforms the SOTA neural decoder (even on unseen codes), (2) The method has a smaller capacity than the SOTA (i.e. fewer parameters), and (3) The method is totally length/rate/code-invariant.

---

> > ### Author Response · Authors · 2023-11-22
> >
> > We thank once more the reviewer for the comprehensive feedback and the ongoing discussion. We hope that we have comprehensively addressed all the remaining questions.
> >
> > Should you have any additional inquiries, please do not hesitate to inform us. If we have satisfactorily addressed your concerns, we would appreciate your consideration in revisiting the score.

---

> > ### Comment · Reviewer_w8i1 · 2023-11-22
> >
> > Thanks authors for your response and the clarification of your simulation error.
> > However, your response regarding the usefulness of the generalist code remains unmoving: 1. In communication, every little compromise of rate (relative to capacity) counts. The foundational Mode is making significant sense in ML because there are other downstream tasks that people then wish to manipulate for their downstream objectives. However, in error control coding, we never accept that one to design a decoder that can decode multiple codes at the cost of rate drop (from capacity). The goal is to get best code and best decoder to have near capacity performance, whether in asymptotic regime or finite length regime (I.e. finite length capacity). The generalist code abandons that goal, unless the authors show that their decoder performs better than any other decoder out there, which is not true! Until, then, I maintain my view that these codes will find no use in reality other than intellectual exercise, which I give that to authors. The focus should be either the code design or decoder design for short length regimes with the best performance. Additionally, I wish to see how much rate you are compromising compare to the best decoders (classic or otherwise) Reed Muller codes in short length to arrive at the same error performance. Can the authors provide the comparison state of art decoder for ReedMuller and show how much performance their generalist decoder compromise on? Thanks!

---

> ### Author Response · Authors · 2023-11-23
>
> We thank the reviewer for the ongoing discussion and for acknowledging, again, the intellectual value of our work. The remaining concern seems to focus not on the potential of the method but on its capability to provide universally superior decoding performance.
> Allow us to clarify our motivation with respect to your remarks.
>
> > and the clarification of your simulation error.
>
> We wish to note we did not have any simulation error, just a typo in the caption of one of the rebuttal figures (“fine-tuned ECCT” instead of  “fine-tuned **F**ECCT”). As mentioned in the previous answer, there is no such thing as “fine-tuned ECCT”, since ECCT is trained from scratch on each code.
>
> >unless the authors show that their decoder performs better than any other decoder out there, which is not true!
> This seems to be the core of the reviewer’s concern and it is exactly what our foundation neural decoder line of work aims to achieve. The main dispute is whether a first work in this direction needs to be the ultimate solution.
>
> We wish to discuss a few points.
>
> 1. Our method obtains state-of-the-art in neural decoding almost always for inference-time zero-shot (remarkably) on codes that are unseen during training, and across all experiments for (i) codes from the training set without further training and (ii) finetuned on unseen codes.
> 2. Previous neural decoders have been shown to outperform or almost match the performance of the best classical decoders (cf. [1,2] for LDPC or BCH codes (BP) and [3] for Polar codes (SCL)).
> 3. Our method, as a machine learning-based solution in general and a neural decoder in particular, may not be the *“best possible universal decoder on every possible code”* ([no free lunch theorem](https://en.wikipedia.org/wiki/No_free_lunch_theorem)).
> Still, our approach contributes significant new scientific advancements in pursuit of this objective, benefiting both the machine learning and error-control coding fields.
> 4. Our state-of-the-art method is the first neural decoder to remain code, rate, and length invariant, enabling the performance and deployment advantages mentioned in the manuscript and the previous answers.
> 5. As previously mentioned, if the universal decoding is not of *practical* interest, our method can be trained or fine-tuned *to decode a few or several families of interest* with high accuracy and potential efficient integration.
> 6. Finally and as mentioned by another reviewer, this is the first time a learned "decoding algorithm rather than a complicated mapping” is proposed, enabling both the analysis of the learned decoding and the potential *differentiable* design of new codes.
>
> > Can the authors provide the comparison state of art decoder for ReedMuller
>
> We provided the performance of our model on six families of codes and many different rates. Reed-Muller codes (related to the Polar codes) are out-of-domain (OOD) for our model as they are not present in our training set, so it is an interesting experiment, especially since *it has not been tested in previous neural decoding works*. Unfortunately, due to time constraints, we will not be able to provide such results within the remaining hours of the discussion phase.
>
> In the context of OOD experiments, please note the OOD zero-shot experiment in Appendix H of the revision, in which the code is 7x the length of the maximum code length in our training set.
>
> [1] *Deep Learning Methods for Improved Decoding of Linear Codes*, Nachmani et al., IEEE Journal of Selected Topics in Signal Processing, 2018
>
> [2] *Error Correction Code Transformer*, Y. Choukroun and L. Wolf, NeurIPS 2022
>
> [3] *Denoising Diffusion Error Correction Codes*, Y. Choukroun and L. Wolf, ICLR 2023

---

### Official Review · Reviewer_dPJy · 2023-11-01

**Soundness:** 3 good
**Presentation:** 2 fair
**Contribution:** 3 good
**Rating:** 6
**Confidence:** 3

**Summary:**

This work embarks on a very ambitious journey towards creating a foundation code for all downstream ECCs. The suggested structure here largely depends on a prior NeurIPS paper ([NeurIPS’22] Choukroun et al., Error Correction Code Transformer, presumably by the same authors) in that it is a specialized Transformer with bitwise embedding and parity-check-matrix-dependent masking, as appropriate for ECC. Albeit similar to prior work, the present proposal contains enough new materials and gives a highly convincing architecture based on code-aware aggregation that depends on the parity-check matrix as well as code-invariant bitwise embedding.

**Strengths:**

The proposed idea is very ambitious, and is based on a highly innovative specialization of the Transformer to the classical problem of decoding received codewords of linear codes. The proposed ideas/strategies are very interesting and convincing (such as bitwise embedding independent of particular codes and incorporation of the parity check matrix in the embedding function). The impact on the field of digital communication and data storage could be large.

**Weaknesses:**

The main issue is that the training of the model is done using codes with lengths up to 150 only, hardly a sufficient length to reflect many modern codes of important applications (testing is also done on relatively small codes, with the largest being a 255 bit BCH). The popular LDPC codes are also curiously missing in the training as well as in the performance evaluation. Likewise for meaningfully long Polar codes. In this sense, I am not sure if the term “foundation model” is justified here. In sum, the idea seems very good, but the validation comes short of a reasonable expectation. I do not feel just saying "we have limited computing resources" would be a good enough excuse for such an ambitious title.

Writings on various parts seem direct copies from [NeurIPS’22] Choukroun et al., Error Correction Code Transformer. Try to differentiate.

**Questions:**

Please respond to the mentioned weaknesses above.

In Tables 1 and 2, the proposed method seem noticeably worse than ECCT on larger codes. Also, In Table 3, ECCT+DM+II gives the best results. Explanations would be  good.

---

> ### Author Response · Authors · 2023-11-14
>
> We thank the reviewer for the supportive review and helpful recommendations.
>
> ## Code Length and Type
> As with most neural decoding works (cf. Related Works section), we focus on short/moderate-length codes for three main reasons.
>
> (i) On very large codes, some decoders (e.g., BP, SC) have been proven to achieve performance close to ML decoding on their respective codes of interest (e.g., LDPC, Polar, respectively), rendering the potential advantage of neural decoding solutions nonexistent/irrelevant.
>
> (ii) The training and/or deployment of the existing neural decoding techniques is not trivial/straightforward for very large codes (i.e., code lengths of thousands of bits).
>
> (iii)  The emergence of applications driven by the Internet of Things created the requirement for optimal decoders of short to moderate codes. For example, 5G Polar codes have code lengths of 32 to 1024 for uplink (UCI) or 128 to 512 for downlink [1].
>
> While the training/testing codes have been randomly selected from the available codes database, as described in the Appendix, we do provide the performance of LDPC (CCSDS)
> https://rptu.de/channel-codes/channel-codes-database/more-ldpc-codes#c94700
>
> Finally, as the reviewer noticed, our current computing resources (described in the paper) cannot allow the efficient training of codes with thousands of bits.
>
> These points are emphasized in the revision.
>
>
> ## Differentiating the Text from ECCT
> As far as we can see, the only part similar to the ECCT is Section 3.1 (now revised to be just the start of Sec 3), which reviews the standard coding setting background. We made several changes to further differentiate this subsection.
>
>
> ## Performance Compared to ECCT on Larger Codes
> We do not observe worse performance on larger codes (e.g., BCH or CCSDS codes), except for the Polar codes only. We believe that this arises from two main reasons. (i) The dataset is highly unbalanced, inducing a low representation of the Polar codes (cf. Section 4.4 paragraph 6, Section 6 paragraph 3, and Appendix C). (ii) Following [2,3,4], we standardized the parity-check matrices of Polar codes (but not of other codes), which may have had an adverse effect on the learned inductive bias.
> That being said, after fine-tuning, ECCT’s performance is outperformed on Polar codes as well, as described in Figure 4.
>
> These points are emphasized in the revision.
>
> ## Table 3 Explanation
>
> We can observe from Table 3 that even while using ECCT masking, invariant initial (II) embedding remains better than using a distinct embedding for each bit (i.e., regular ECCT), where the bit position is taken into account by the FC layer. However, we can observe that using invariant output (IO) coupled with II loses positional information, inducing a drop in accuracy.
>
> A similar effect appears also for ECCT+DM+II which improves over ECCT’s original initial embedding. The conclusion is that ECCT is worse than FECCT in all of its components: initial encoding, masking, and final encoding.
>
> We note here that, as described in the caption of Table 3, ECCT has a larger capacity which makes an exact comparison difficult.
>
> These points are emphasized in the revision.
>
> ## References
> [1]  ETSI 3GPP TS 38.212 “5G NR Multiplexing and channel coding”, v.16.5.0, 2021-03.
>
> [2] Error Correction Code Transformer, Y. Choukroun and L. Wolf, NeurIPS 2022
>
> [3] https://github.com/yoniLc/ECCT
>
> [4] How to Mask in Error Correction Code Transformer: Systematic and Double Masking, S. Park et al., arXiv:2308.08128

---

> ### Author Response · Authors · 2023-11-21
>
> We thank again the reviewer for the valuable feedback. We hope that we have comprehensively addressed all the raised questions and remarks.
>
> Should you have any additional inquiries, please do not hesitate to inform us. If we have satisfactorily addressed your concerns, we would appreciate your consideration in revisiting the score.

---

> > ### Comment · Reviewer_dPJy · 2023-11-22
> >
> > Thanks for the response, but it appears that the authors are saying that since there already exist good decoders for very large codes, there is no room for a foundational decoder there. Isn't the motivation for a foundation decoder to have a single decoder design that can handle decoding of any codes (or a wide variety of practical codes) with only a little tweaking (finetuning), without having to design separate decoders for different codes? For any specific short or moderate length code, there already are reasonably good practical decoders that are in use today. So following your argument, I do not see a need for a neural decoder solution thetre as well. You also say the training and/or deploying existing neural decoding techniques is not trivial/straightforward for very large codes (i.e., code lengths of thousands of bits). So you were not able to valiadate your method for large codes. Shouldn't you then revise your title to A Foundation Model for Short (or Moderate Length) Error Correction Codes?
> >
> > Overall, all my concerns remain the same after the author rebuttal, and I am atually tempted to lower my score. Nevertheless, I still feel that the approach is novel and promissing, so I will resist my temptation and keep my score as is.

---

> > > ### Author Response · Authors · 2023-11-22
> > >
> > > We are sorry that our answer did not address the code length concerns. We believe there is a misunderstanding arising from our explanation regarding neural decoders in general.
> > >
> > > > it appears that the authors are saying that since there already exist good decoders for very large codes, there is no room for a foundational decoder there.
> > >  Isn't the motivation for a foundation decoder to have a single decoder design that can handle decoding of any codes (or a wide variety of practical codes) with only a little tweaking (finetuning), without having to design separate decoders for different codes?
> > >
> > > We are sorry we were not clear enough, allow us to clarify our intended meaning. As the reviewer rightly wrote, *our foundation model should and can, by design, support any code of any rate/length*. What we meant was that similar to every neural decoding work (since the seminal work of [1]), we provide performance analysis on shorter codes which are the codes of interest since 1) no substantial accuracy improvement can be obtained on large capacity reaching codes and 2) modern applications focus on shorter codes.
> > >
> > > Following the reviewer's concern, we provide the performance of our pretrained model on a **zero-shot task** on a large BCH(1023,1013) code. This is seven times the size of the largest code in our training set, and 13 times the mean code length in the training set. The result is presented [here](https://ibb.co/P14N7hB) . Baselines are taken from [here](https://rptu.de/channel-codes/ml-simulation-results#&gid=lightbox-group-62136&pid=4)
> > >
> > > As can be observed, the model we trained for short codes is capable of generalizing to unseen codes with lengths larger by more than one order of magnitude than the codes the model has been trained on. This is strong support for the length invariance and generalization capabilities of the input, masking, and code-aware prediction modules. We thank the reviewer for suggesting we address the domain of larger codes. The result has been appended to Appendix H.
> > >
> > > >You also say the training and/or deploying existing neural decoding techniques is not trivial/straightforward for very large codes (i.e., code lengths of thousands of bits). So you were not able to validate your method for large codes.
> > >
> > > As stated before our model can support any code of any length. What we meant there was that if, for instance, you wish to train our foundational model on larger codes, the complexity obviously increases with the code length and would require longer training and validation time, as well as larger resources/GPUs in order to maintain a similar training setting, such as the batch size.
> > >
> > > We hope we shed some light on our previous answers and that we were able to address your concerns.
> > >
> > >
> > > [1] *Deep Learning Methods for Improved Decoding of Linear Codes*, Nachmani et al., IEEE Journal of Selected Topics in Signal Processing, 2018

---

> > > > ### Comment · Reviewer_dPJy · 2023-11-22
> > > >
> > > > I thank the authors for clarification and additional validation on a large BCH code. I remain positive and will recommend acceptance.

---

### Official Review · Reviewer_1rnF · 2023-11-04

**Soundness:** 4 excellent
**Presentation:** 4 excellent
**Contribution:** 4 excellent
**Rating:** 8
**Confidence:** 5

**Summary:**

The authors proposed a model and is trained on multiple codes and can then be applied to an
unseen code.
Transformer architecture in multipleways:
(1) a code-invariant initial embedding, which is also position- and lengthinvariant,
(2) a learned modulation of the attention maps that is conditioned on the Tanner graph
(3) a length-invariant code-aware noise prediction module that is based on the parity-check matrix

**Strengths:**

1.Error control coding implemtation on the deep learning technique is highly encourgaing.
2.Authors got the optimized Results in terms of BER.

**Weaknesses:**

1.Conclusion should be rewritten based on the results presented mentioning the future scope.

**Questions:**

1.Conclusion should be rewritten based on the results presented mentioning the future scope.

---

> ### Author Response · Authors · 2023-11-14
>
> We thank the reviewer for the very supportive review.
>
> The conclusion is rewritten and includes the future scopes of research as follows.
>
> *“Next steps should focus on the deployment of the method on existing error correction embedded systems via the utilization of Transformer acceleration methods along with extensive tuning and training of the codes and channels of interest. Additionally, the definition of a universal differentiable neural decoder may open the door to the optimization of codes and even to the learning of new families of codes.”*

---

### Meta-Review · Area_Chair_SVSK · 2023-12-09

**Metareview:**

**Summary**

This paper proposes a "foundational" model, named FECCT, for decoding binary linear error-correcting codes. The proposed model is built on the existing model ECCT with several modifications, such as a code-invariant initial embedding (Section 4.1), Tanner graph distance masking (Section 4.2), and parity-check aware prediction (Section 4.3), and is able to perform decoding of codes unseen during training.

**Strengthes**

The proposal serves as a proof-of-concept study of the feasibility of transformer-based universal decoding of binary linear codes, and is expected to stimulate further studies along the direction.

**Weaknesses**

I share exactly the same concern with Reviewer w8i1 on practical usefulness of the proposal: Advantage of the proposal might be questionable for longer codes because of heavy computational burden for training as well as availability of capacity-achieving strategies, and also for moderate-length codes for which dedicated decoders are often available. I would, however, weigh the above-mentioned strength.

Upon my own reading of this paper, I noticed several minor points that would need revision, as listed in the following.
- Page 3, lines 14, 16, 18: The operator $\cdot$ appearing in the formulas should be the Hadamard product $\odot$.
- Page 3, line 34: The size of the argument $H$ of $g(H)$ is not $(n-k)\times k$ but $(n-k)\times n$, as it is the parity-check matrix.
- Page 3, line 39: On the right-hand side of equation (5), there seems to be a size mismatch. My guess is that $W_o$ is not of size $(n-k)\times n$ but of size $n\times (2n-k)$, and that the right-hand side of equation (5) should read  $W_o\Phi W_{d\to1}$.
- Page 4, line 31: $G(H)$ $\to$ $\mathcal{G}(H)$
- Page 4, lines 34-35: In equation (5) $\psi(\mathcal{G}(H))$ is **multiplied** elementwise to $QK^T$ **outside** the softmax, whereas in equation (4) $g(H)$ is **added** to $QK^T$ **inside** the softmax, so that one cannot simply say that the latter is a special case of the former.
- Page 4, Figure 1 (d): Because the Tanner graph is bipartite, the variable-variable block and the check-check block of the distance matrix should consist of even numbers only, while the variable-check block should consist of odd numbers only. One observes in Figure 1 (d) that it is not the case. One also expects that the elements in the variable-check block should be equal to 1 if the corresponding elements of the parity-check matrix are 1, whereas one cannot find such a structure in the variable-check block of the distance matrix shown in Figure 1 (d).  These apparent inconsistencies should be revisited and corrected appropriately. Also, using a color gradient to represent integer values, as shown in the color bar on the right-hand side, is confusing.
- Page 5, line 28: In equation (8) we observe similar size inconsistencies to those mentioned above on equation (5).
- Page 6, line 4: $1\ldots 10$ $\to$ $1,\ldots,10$
- Page 6, line 12: 2K $\to$ $2K$
- Page 6, lines 16, 17, 19: The vector-vector multiplication in these formulas should be the Hadamard product, which should be denoted by $\otimes$. $x_s^2$ should be written as $x_s^{\otimes 2}$.
- Page 7, line 20: $Eb/N_0$ $\to$ $E_b/N_0$

**Justification For Why Not Higher Score:**

This paper would be of interest to researchers in the interdisciplinary fields of coding theory and machine learning, whereas it would not be of interest to a wider range of researchers in representation learning.

**Justification For Why Not Lower Score:**

Although one might be able to argue that practical usefulness of the proposal is questionable, I still believe that the proposal is novel and potentially valuable in that it will stimulate further studies along the direction.

---

### Decision · Program_Chairs · 2024-01-16

Accept (poster)